# Slo1 is the principal potassium channel of human spermatozoa

Nadja Mannowetz[1], Natasha M Naidoo[1†], Seung-A Sara Choo[1†], James F Smith[2], Polina V Lishko[1*]

[1]Department of Molecular and Cell Biology, University of California, Berkeley, Berkeley, United States; [2]Department of Urology, University of California, San Francisco, San Francisco, United States

**Abstract** Mammalian spermatozoa gain competence to fertilize an oocyte as they travel through the female reproductive tract. This process is accompanied by an elevation of sperm intracellular calcium and a membrane hyperpolarization. The latter is evoked by $K^+$ efflux; however, the molecular identity of the potassium channel of human spermatozoa (hKSper) is unknown. Here, we characterize hKSper, reporting that it is regulated by intracellular calcium but is insensitive to intracellular alkalinization. We also show that human KSper is inhibited by charybdotoxin, iberiotoxin, and paxilline, while mouse KSper is insensitive to these compounds. Such unique properties suggest that the Slo1 ion channel is the molecular determinant for hKSper. We show that Slo1 is localized to the sperm flagellum and is inhibited by progesterone. Inhibition of hKSper by progesterone may depolarize the spermatozoon to open the calcium channel CatSper, thus raising $[Ca^{2+}]$ to produce hyperactivation and allowing sperm to fertilize an oocyte.

*For correspondence: lishko@berkeley.edu

†These authors contributed equally to this work

Competing interests: The authors declare that no competing interests exist.

## Introduction

Mammalian spermatozoa are unable to fertilize the oocyte immediately after their deposit into the female reproductive tract. Instead, they have to undergo a final maturation termed capacitation, during which spermatozoa gain competence to fertilize (*Chang, 1951*; *Austin, 1952*). Early stages of capacitation include the bicarbonate-mediated acceleration of sperm beat frequency and an increase in linear motility (*Visconti et al., 1995a*, *1995b*, *1999*, *2002*; *Chen et al., 2000*; *Wennemuth et al., 2003*; *Wandernoth et al., 2010*; *Mannowetz et al., 2011*). Late stages of capacitation comprise—amongst others—intracellular alkalinization (*Meizel and Deamer, 1978*), elevation of intracellular $Ca^{2+}$ (*Visconti et al., 2002*), and membrane hyperpolarization (*Zeng et al., 1995*; *Arnoult et al., 1996*; *Demarco et al., 2003*). These interdependent processes are regulated by sperm ion channels, of which Hv1 and CatSper (Cation channel of sperm) were identified as the major $H^+$ and $Ca^{2+}$ channels of human spermatozoa (*Ren et al., 2001*; *Kirichok et al., 2006*; *Lishko and Kirichok, 2010*; *Lishko et al., 2010*, *2011*; *Ren and Xia, 2010*; *Strunker et al., 2011*; *Lishko et al., 2012*). However, the identity of the principal human $K^+$ channel remained elusive.

Potassium channels are indispensable for normal sperm physiology, since they regulate membrane potential and cell motility. Recently, an alkalinization-sensitive sperm $K^+$ channel, encoded by the *kcnu1* gene (also known as *Slo3*), was shown to be essential for male fertility in mice (*Schreiber et al., 1998*; *Navarro et al., 2007*; *Santi et al., 2010*; *Zeng et al., 2011*). It has been assumed, but never proven, that the $K^+$ channel of human sperm has a similar molecular identity. The *Slo* gene family is represented by *Slo1*, *Slo2*, and *Slo3* (*Wei et al., 2005*). These channels possess seven transmembrane helices S0–S6, with the S1–S6 helices exhibiting homology to classic voltage-gated $K^+$ channels. They are tetramers of α subunits, with the $K^+$-selective pore formed by S5 and S6 (*Adelman et al., 1992*; *Butler et al., 1993*; *Diaz et al., 1998*; *Cui and Aldrich, 2000*). In addition, the Slo1 channel contains

**eLife digest** The sperm cells that are released into the female reproductive tract when a mammal ejaculates, are not capable of fertilizing an egg right away, so they must go through a process called maturation. The early stages of this process involve interactions with the seminal fluid that increase the motility of the sperm cells, and the latter stages involve interactions with the walls of the reproductive tract and vaginal secretions to ensure that the sperm cells move toward the egg. Many of these interactions involve positive ions entering and leaving the sperm cells via ion channels.

The properties of the ion channels that allow protons and calcium ions to move into and out of human sperm cells are well understood, but little is known about the channels that control the movement of the potassium (K) ions. At first it was assumed that the molecular structure of these channels was similar to that of the Slo3 potassium channel in mouse sperm, but crucial differences between human and mouse sperm cells have been reported in recent years.

Now Mannowetz et al. have shown that the potassium channel in human sperm is opened by increased levels of calcium ions inside the sperm cells. Moreover, the pH inside the sperm cells had no influence on this process. Furthermore, the channel was blocked by three toxins that have no effect on the Slo3 potassium channels in mice, but are known to block a type of potassium channel known as Slo1. Mannowetz et al. then used a technique called Western blotting to confirm the presence of Slo1 potassium channels in the tails of human sperm cells.

Mannowetz et al. also showed that the Slo1 potassium channel can be blocked by the female hormone progesterone. This is important because blocking the potassium channels causes the calcium ion channels in the cells to open fully, and the resulting influx of calcium ions triggers a process called sperm hyperactivation that makes it possible for the sperm cell to fertilize the egg. By clearly showing the fundamental differences between human sperm cells and mouse sperm cells, this work stresses the need to exercise caution in using mice as a model of male fertility in humans.

a large cytosolic C-terminus with two intracellular regulators of $K^+$ conductance (RCK), both of which contain high affinity $Ca^{2+}$ binding sites (*Jiang et al., 2001*; *Yuan et al., 2010*). These structural elements give Slo1 channels the ability to sense changes in both voltage and intracellular $Ca^{2+}$ concentrations (*Marty, 1981*; *Pallotta et al., 1981*; *Barrett et al., 1982*; *Latorre et al., 1982*; *Schreiber et al., 1999*). Due to their large single-channel conductance of 60–270 pS, Slo1 channels are also known as big potassium (BK) or maxi K channels (*Atkinson et al., 1991*; *Kaczorowski et al., 1996*; *Salkoff et al., 2006*). Slo3 channels, on the other hand, lack the $Ca^{2+}$ bowl (*Schreiber et al., 1999*; *Xia et al., 2004*), but are sensitive to intracellular alkalinization (*Schreiber et al., 1998*; *Zhang et al., 2006a*, *2006b*; *Navarro et al., 2007*). The pore-forming α subunits of Slo channels are associated with auxiliary β- and γ-subunits (*Behrens et al., 2000*; *Brenner et al., 2000*; *Uebele et al., 2000*; *Yan and Aldrich, 2010*, *2012*; *Yang et al., 2011*), which interact with the S0 segment of the α subunit. Several studies demonstrate that the association with different subunits impacts channel pharmacological and gating properties. In addition, splice variants of the *Slo1* mRNA contribute to the functional diversity of BK channels (*Fodor and Aldrich, 2009*; *Johnson et al., 2011*). Apart from responding to different stimuli, Slo1 and Slo3 channels are distributed discretely within the body as shown in numerous animal studies. Slo1 is detectable in excitable tissues, such as in hippocampus (*Hicks and Marrion, 1998*), smooth muscle cells (*Knaus et al., 1994a*, *1994b*) and adrenal chromaffin cells (*Solaro and Lingle, 1992*), whereas Slo3 transcripts are exclusively expressed in male germ cells (*Schreiber et al., 1998*). Male $Slo1^{-/-}$ animals are able to produce offspring when paired with $Slo^{+/+}$ females. However, the litter size was normal only in 10% of the matings (*Meredith et al., 2004*). Abolishing the Slo3 gene results in more dramatic changes in testicular spermatozoa, such as morphological abnormalities after capacitation, reduced progressive motility, impaired acrosome reaction, and membrane depolarization during capacitation (*Schreiber et al., 1998*; *Santi et al., 2010*; *Zeng et al., 2011*). These data indicate that Slo channels are essential for male fertility in mice, which makes them possible candidates for being the major $K^+$ channel of human sperm.

The goal of our work was to resolve the identity of the major $K^+$ channel in human ejaculated spermatozoa. By applying the patch-clamp technique to ejaculated and epidydymal human sperm cells, we found that human $K^+$ currents are insensitive to intracellular alkalinization but are dependent

on intracellular $[Ca^{2+}]$. We furthermore demonstrate that the human sperm potassium (hKSper) current is inhibited by three known Slo1 channel inhibitors: charybdotoxin (*Anderson et al., 1988*; *MacKinnon and Miller, 1988*), iberiotoxin (*Galvez et al., 1990*; *Candia et al., 1992*; *Giangiacomo et al., 1992*) and paxilline (*Knaus et al., 1994c*; *Sanchez and McManus, 1996*; *Zhou et al., 2010*), as well as by the micromolar concentrations of progesterone. Taking together our electrophysiological, biochemical, and immunocytochemistry data, we conclude that the Slo1 protein constitutes a major potassium channel of human spermatozoa. Therefore, the molecular identity of human KSper is distinct from that of murine KSper, which is represented by the Slo3 protein.

## Results

### hKSper currents originate from the sperm flagellum

The flagellar pH-dependent $Ca^{2+}$ channel CatSper is indispensable for male fertility. However, to gain its full activity several events must be met: intracellular alkalinization, presence of progesterone and membrane depolarization (*Ren et al., 2001*; *Kirichok et al., 2006*; *Lishko and Kirichok, 2010*; *Lishko et al., 2011*; *Strunker et al., 2011*). Since $K^+$ channels are involved in the regulation of membrane potential, we hypothesized that the human KSper current ($I_{KSper}$) also originates from the sperm tail to support CatSper activity. To address this question, we recorded from both whole sperm cells and isolated sperm flagella (*Figure 1*). To isolate $I_{KSper}$ from $I_{CatSper}$, we recorded $K^+$ currents in a potassium methanesulfonate-based solution in the presence of 0.1–1 mM extracellular $Ca^{2+}$. When divalent cations are absent from the extracellular solution, so called divalent free (DVF) condition, CatSper is able to conduct monovalent ions, such as $K^+$. However, in the presence of 0.1–1 mM external $Ca^{2+}$, $I_{CatSper}$ is effectively blocked (*Kirichok et al., 2006*; *Lishko et al., 2011*; *Smith et al., 2013*), thus leading to pure $K^+$ conductance through $K^+$ channels. As shown in *Figure 1A,B*, $K^+$ currents elicited under DVF conditions were approximately four times larger than the current recorded in the presence of 1 mM $Ca^{2+}$. The larger potassium currents (gray bars) represent a mixture of the $K^+$ efflux through CatSper

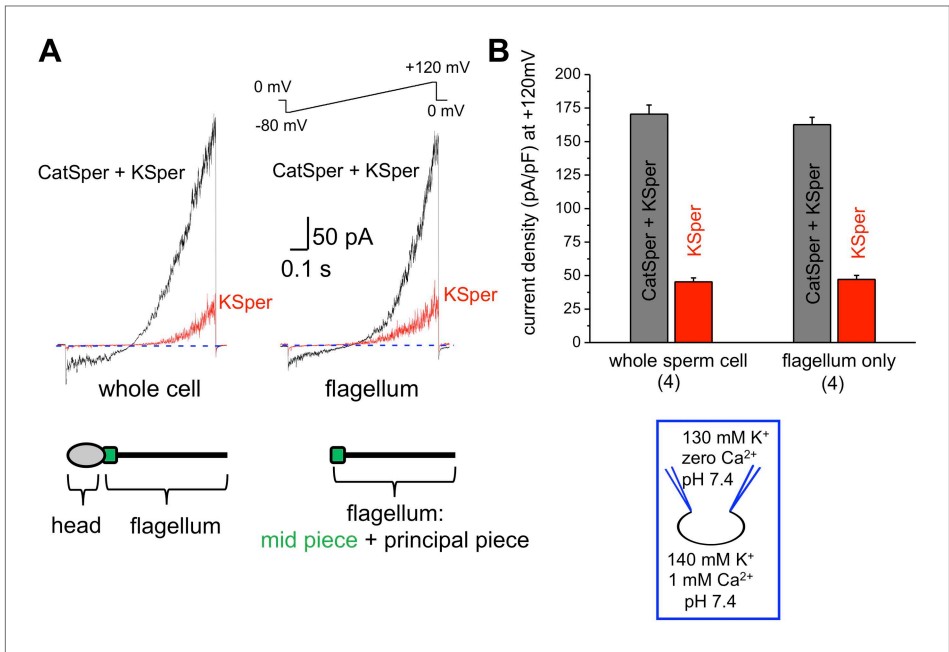

**Figure 1**. hKSper currents originate from the sperm tail. (**A**) $I_{KSper}$ was recorded in response to a voltage ramp as shown. Shown are representative traces from whole spermatozoon (left panel; recordings are from the same cell) and sperm tail (right panel; recordings are from the same flagellum). Black traces represent currents in divalent free conditions, which allow $K^+$ current through CatSper. Red traces show true $I_{KSper}$. Latter was recorded in the presence of 1 mM extracellular $Ca^{2+}$, which inhibits monovalent currents through CatSper. (**B**) Current densities were obtained at +120 mV and presented as mean ± SEM. (n), number of experiments. Four different sperm cells (or four different sperm flagella) of two different human donors were used.

and KSper while only KSper current remains in the presence of external calcium (red bars). Similar amplitudes of KSper currents recorded from whole sperm cells or sperm flagella indicate that $I_{KSper}$ originate primarily from the sperm flagellum, in the same manner as does $I_{CatSper}$ (**Figure 1B**; **Lishko et al., 2011**).

## hKSper currents are insensitive to intracellular alkalinization

Sperm intracellular alkalinization was shown to be essential for murine KSper (Slo3) activation (**Navarro et al., 2007**; **Santi et al., 2010**; **Zeng et al., 2011**). Recently, recombinant human Slo3 co-expressed with a γ-subunit was also shown to exhibit pH-dependency (**Leonetti et al., 2012**). However, its pH-sensitivity was shifted toward a more acidic pH range than that of mouse Slo3. Therefore, we decided to test whether human KSper exhibits the same pH sensitivity and recorded K$^+$ currents from human spermatozoa under conditions when intracellular pH (pH$_i$) was held either at pH 7.4 or 5.5, and external pH kept at 7.4. To evoke intracellular alkalinization, 10 mM of NH$_4$Cl was added to the external (bath) solution, which is a standard technique to effectively and quickly raise an intracellular pH (**Babcock et al., 1983**; **Kirichok et al., 2006**; **Navarro et al., 2007**). In the experiments with a pH$_i$ of 5.5, 1 mM of Zn$^{2+}$ was added to the bath solution to inhibit sperm voltage-gated proton channel (Hv1) activity (**Lishko et al., 2010**). As shown in **Figure 2A**, human $I_{KSper}$ remained unaffected by intracellular alkalinization at both pH$_i$ 7.4 (upper left panel) and pH$_i$ 5.5 (lower left panel). However, K$^+$ currents were greatly potentiated by intracellular alkalinization in the absence of divalent cations (**Figure 2A,B**, right panels), which was primarily due to the activation of pH-dependent K$^+$ efflux through CatSper channels. Note that in DVF conditions, control currents were larger than control KSper currents, due to the efflux of potassium ions through both KSper and CatSper channels. Intracellular alkalinization up-regulates CatSper channel activity, therefore increasing the potassium efflux through it, while KSper currents remain unchanged.

To exclude that components of the seminal plasma may alter human sperm K$^+$ channel behavior with regard to pH sensitivity, we also recorded K$^+$ currents from human epididymal spermatozoa from

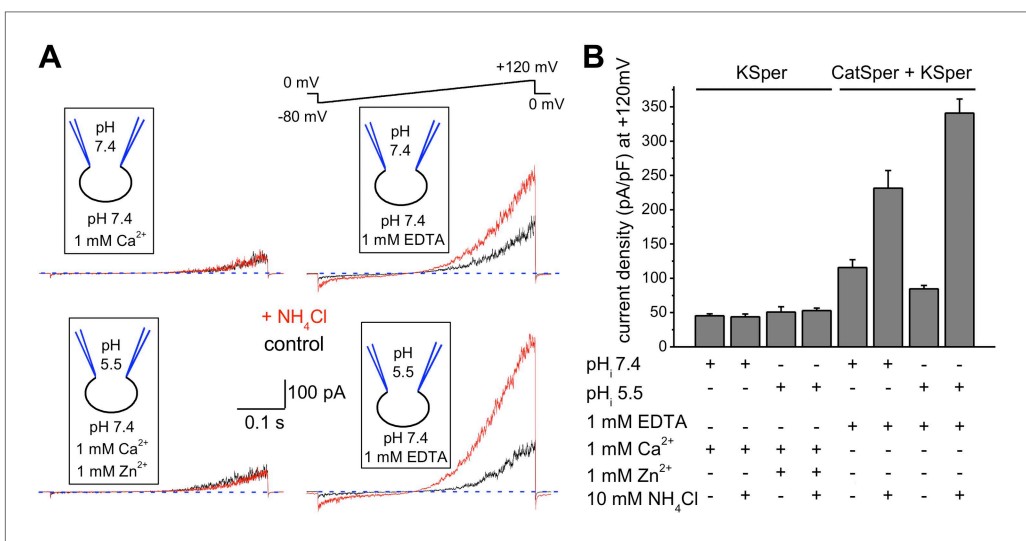

**Figure 2**. hKSper currents are insensitive to intracellular alkalinization. (**A**) Representative KSper currents were recorded from sperm cells in response to voltage ramps as shown. Recordings were done with various pH$_i$ as indicated. The bath solution containing 1 mM Ca$^{2+}$ was used to inhibit K$^+$ current through CatSper (left panels). Right panels show traces in divalent free conditions, which allow K$^+$ current through CatSper. Intracellular alkalinization was evoked by addition of 10 mM NH$_4$Cl to the bath (red traces). A weak intracellular buffer (5 mM of HEPES or MES) allowed instantaneous pH changes. Zn$^{2+}$ was used to block H$^+$ currents via Hv1 at acidic intracellular pH. The upper panels and the lower panels are recordings from two different sperm cells. (**B**) KSper and CatSper/KSper current densities (CDs) recorded from sperm cells as shown in (**A**). At pH$_i$ 7.4 KSper CDs were: 45 ± 3 pA/pF (control) and 44 ± 4 pA/pF (plus NH$_4$Cl). These values were similar at pH$_i$ 5.5: CDs were: 51 ± 8 pA/pF (control) and 53 ± 4 pA/pF (plus NH$_4$Cl). However, under DVF conditions that permit K$^+$ efflux through CatSper, CDs at pH$_i$ 7.4 were: 116 ± 11 pA/pF (control) and 231 ± 26 pA/pF (plus NH$_4$Cl). At pH$_i$ 5.5, CDs were: 85 ± 5 pA/pF (control) and 341 ± 21 pA/pF (plus NH$_4$Cl). Shown are CDs acquired at +120 mV and presented as mean ± SEM; n = 4–6 independent experiments with cells from four different human donors.

a fertile patient undergoing vasectomy reversal. As shown in *Figure 3*, $I_{KSper}$ did not change upon intracellular alkalinization. This indicates that the lack of hKSper pH-sensitivity is not due to the effect of seminal plasma, but rather it is an intrinsic property of the human sperm potassium channel. This particular feature of human KSper differentiates it from mouse KSper and suggests that the molecular identities of the channels are different.

## hKSper is activated by intracellular calcium

As mentioned earlier, the pH-dependent mouse KSper channel is encoded by the *Slo3* gene, while other members of the *Slo* family, such as Slo1 are not pH-dependent, but rather $Ca^{2+}$-dependent. To determine if intracellular calcium affects human $I_{KSper}$, we recorded $K^+$ currents under different intracellular free $Ca^{2+}$ concentrations: 0, 0.1 or 50 μM (*Figure 4*) with 0.1 mM $Ca^{2+}$ in the bath solution. As illustrated in *Figure 4A,B*, the outward $I_{KSper}$ was slightly increased with $[Ca^{2+}]_i$ = 0.1 μM compared to the control (zero calcium). Under these conditions, $I_{KSper}$ exhibited outward rectification. However, with a $[Ca^{2+}]_i$ = 50 μM, not only was the outward current potentiated twofold, but an inward potassium current was also present. Interestingly, intracellular calcium also notably decreased the activation time for human KSper (*Figure 4A*, lower panel). However, the quantitative measurements of activation time constant in the presence of calcium were hindered by a fast channel kinetics that overlapped with capacitance artifacts.

Regulation by $Ca^{2+}$ is a hallmark behavior of Slo1, but not Slo3 channels. Our results indicate that intracellular $Ca^{2+}$, and not pH, is a driving force for the opening of the human KSper channel and suggest that the molecular identity of hKSper might be the Slo1 protein rather than Slo3.

## hKSper is sensitive to Slo1 channel blockers charybdotoxin, iberiotoxin, and paxilline

To verify the molecular identity of human KSper, we applied three of the known Slo1 channel blockers to the bath solution: charybdotoxin (ChTX) (*Anderson et al., 1988*; *MacKinnon and Miller, 1988*), iberiotoxin (IbTX) (*Galvez et al., 1990*; *Candia et al., 1992*; *Giangiacomo et al., 1992*), and paxilline (*Knaus et al., 1994c*; *Sanchez and McManus, 1996*; *Zhou et al., 2010*). *Figure 5A,B* shows a potent and reversible inhibition of human $I_{KSper}$ by 93% in the presence of 1 μM ChTX. Human $K^+$ currents were also effectively blocked by both 100 nM IbTX (*Figure 6A,B*) and 100 nM paxilline (*Figure 7A,B*) with 87% and 62% inhibition, respectively. To verify that this pharmacological profile was specific to Slo1, we also recorded $K^+$ currents from mouse sperm, which express Slo3 (*Schreiber et al., 1998*; *Zhang et al., 2006a*; *Navarro et al., 2007*; *Santi et al., 2010*; *Zeng et al., 2011*). It was previously reported that mouse Slo3 is insensitive to ChTX, IbTX, and paxilline (*Tang et al., 2010*), and indeed *Figures 5C,D, 6C,D, and 7C,D* demonstrate that mouse $K^+$ currents remained unaffected upon stimulation with 1 μM ChTX, 100 nM IbTX, or 500 nM paxilline. The fact that human, but not mouse, KSper is sensitive to Slo1-specific channel blockers strongly suggests that Slo1 forms the potassium channel in human sperm.

Mouse KSper appeared to have both notably larger current amplitudes and current densities (*Figures 5–7*). Mouse spermatozoa are twice larger than human sperm cells: human sperm capacitance is usually within 1 pF, while the capacitance of mouse sperm is about 2.5 pF (*Kirichok et al., 2006*; *Lishko et al., 2010, 2011*). However, the fact that KSper current densities (pA/pF) are still

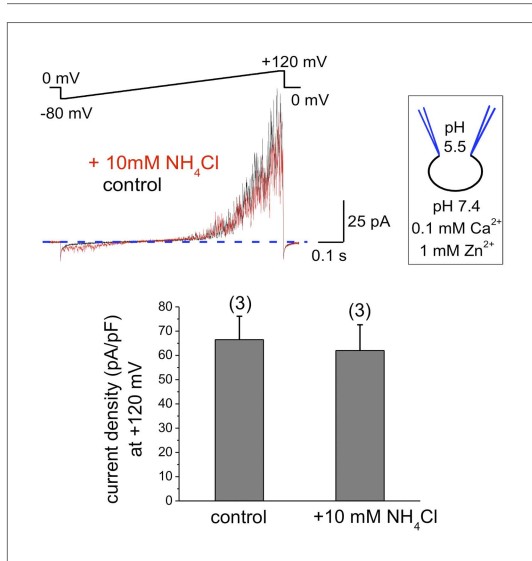

**Figure 3**. hKSper currents from human epididymal spermatozoa are insensitive to intracellular alkalinization. The upper panel shows representative $I_{KSper}$ traces recorded from human epididymal spermatozoa (whole sperm cell) in the control (black) and in the presence of 10 mM $NH_4Cl$ (red). The lower panel presents mean currents acquired at +120 mV; (n), number of experiments. $I_{KSper}$ did not change upon intracellular alkalinization with current densities averaging at 67 ± 10 pA (control) and 62 ± 11 pA (after addition of 10 mM $NH_4Cl$). Three epididymal spermatozoa were tested.

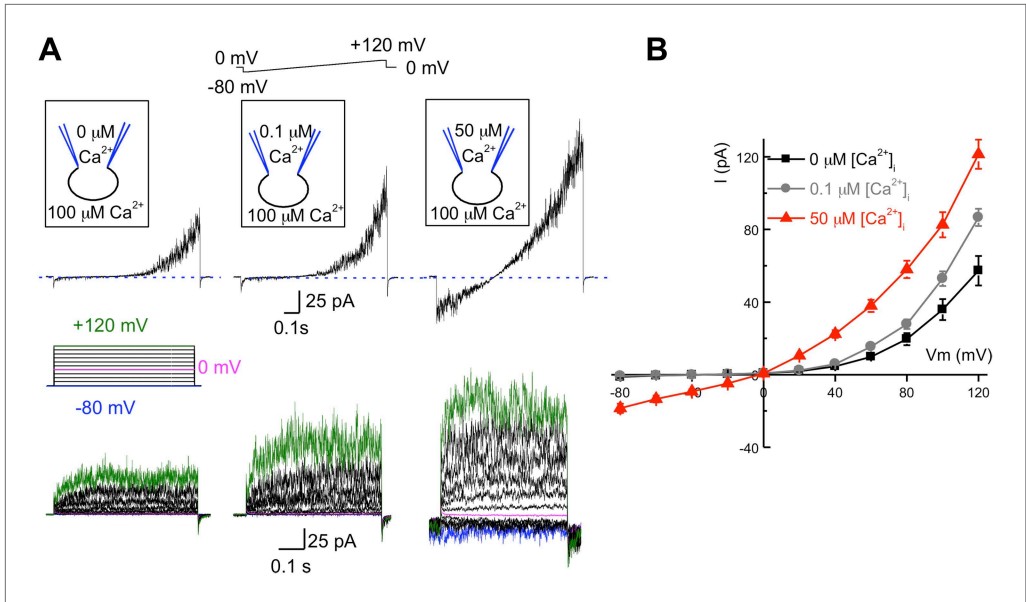

**Figure 4**. hKSper is activated by intracellular calcium. (**A**) Upper panels: representative $I_{KSper}$ recorded with various intracellular $[Ca^{2+}]_{free}$ as indicated, in response to a voltage ramp. Lower panels: corresponding representative $I_{KSper}$ elicited by a step protocol from a holding potential of −80 mV to +120 mV with 20 mV increments. For clarity, traces at −80 mV, 0 mV, and +120 mV are labeled in blue, magenta, and green, respectively. Representative traces were obtained from three different sperm cells (upper and lower panels). (**B**) Current–voltage (I–V) relationship in response to 0 μM (black), 0.1 μM (gray), and 50 μM (red) intracellular $[Ca^{2+}]_{free}$. At a membrane potential (Vm) of −80 mV, potassium currents were: -1.2 ± 0.5 pA ($[Ca^{2+}]_i$ = 0), -0.8 ± 0.2 pA ($[Ca^{2+}]_i$ = 0.1 μM), and -18.5 ± 2.6 pA ($[Ca^{2+}]_i$ = 50 μM). At Vm = +120 mV, $I_{KSpers}$ were 57 ± 8 pA ($[Ca^{2+}]_i$ = 0), 87 ± 5 pA ($[Ca^{2+}]_i$ = 0.1 μM), and 122 ± 8 pA ($[Ca^{2+}]_i$ = 50 μM). Data are shown as means ± SEM; n = 6–11 independent experiments with cells from six different donors. Data are from whole sperm cells.

larger in mouse sperm than in human spermatozoa indicates the potential differences in KSper expression and distribution along the sperm flagella.

## hKSper is blocked by progesterone

We and others previously have shown that the sperm-specific calcium channel CatSper is activated by progesterone (*Lishko et al., 2011*; *Strunker et al., 2011*). Progesterone (P) shifts CatSper activation to more physiological, hyperpolarized, membrane potentials (*Lishko et al., 2011*). To test whether progesterone has any effect onto hKSper, we recorded $I_{KSper}$ in the presence of different progesterone concentrations in the bath solution. *Figure 8A,C* shows that hKSper outward currents were blocked by progesterone in a dose-dependent manner. We have determined that progesterone's half-maximum inhibitory concentration (IC$_{50}$) for hKSper is 7.5 ± 1.3 μM (*Figure 8B*). Moreover, mouse Slo3 turned out to be insensitive to 10 μM of progesterone (*Figure 9*), which is above the IC$_{50}$ for human KSper. Since potassium channels are well known to regulate the membrane potential (*Navarro et al., 2007*), it is likely that the inhibition of human KSper by P will produce membrane depolarization and create favorable conditions for opening of CatSper. CatSper activation, in turn, will result in an elevation of intracellular $[Ca^{2+}]$ and trigger hyperactivated motility. To test this hypothesis we selectively blocked human KSper by adding 100 nM ChTX to high saline (HS) bath solution in which sperm cells are usually kept, and recorded any changes in sperm motility. As evident from *Video 1* sperm motility was symmetrical in the absence of the Slo1 inhibitor (ChTX). However, incubation in 100 nM ChTX for 25 min resulted in sperm cells exhibiting an asymmetrical motility pattern similar to hyperactivation (*Video 2*). The normal, symmetrical motility was resumed after a prolonged washout (data not shown).

## The Slo1 protein present in human spermatozoa

To confirm that the Slo1 protein is actually present in human spermatozoa, we performed immunostaining with anti-Slo1 specific antibodies. *Figure 10A* demonstrates that the antibody selectively stained the

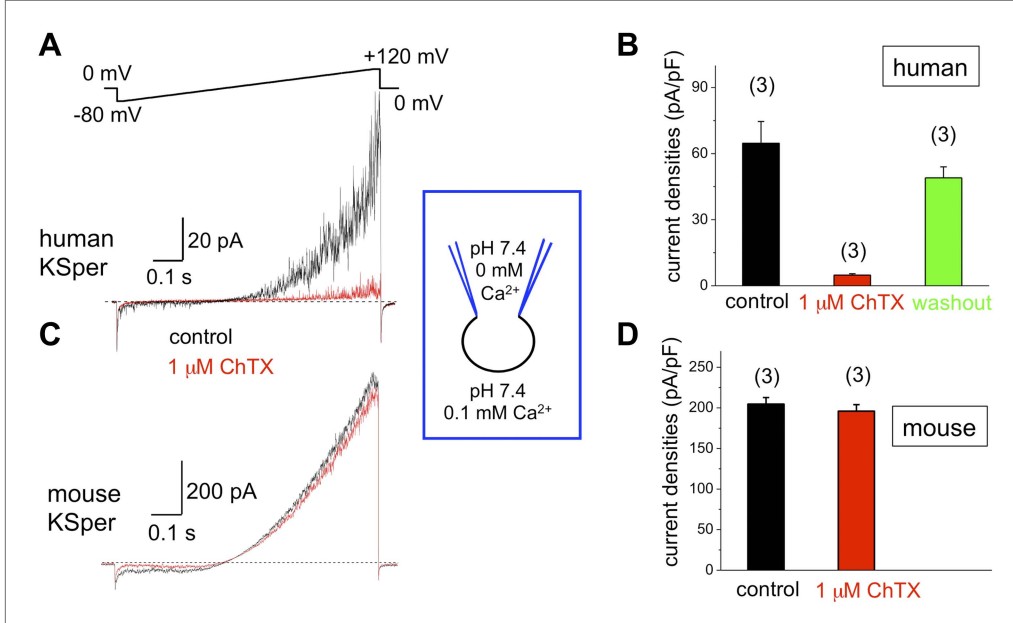

**Figure 5**. Human, but not mouse KSper is sensitive to the Slo1 channel blocker charybdotoxin (ChTX). (**A**) Representative human $I_{KSper}$ traces under control conditions (black) and in the presence of 1 μM ChTX (red) elicited in response to the given voltage ramp. (**B**) Mean current densities (CDs) ± SEM calculated at +120 mV. CDs (human) were: 65 ± 10 pA/pF (control), 5 ± 1 pA/pF (ChTX), and 49 ± 5 pA/pF (washout). (**C**) Representative mouse $I_{KSper}$ traces under control conditions (black) and in the presence of 1 μM ChTX (red) elicited in response to the voltage ramp as shown in (**A**). (**D**) CDs (mouse) were: 205 ± 8 pA/pF (control) vs 196 ± 8 pA/pF (ChTX). (n), number of experiments. Three human and three mouse sperm cells were used.

principal piece of the sperm flagellum, the same compartment where other sperm ion channels, such as Hv1 and CatSper, reside. The head and the flagellar midpiece region showed no signals (**Figure 10A**, middle and left panel). Furthermore, the presence of the Slo1 protein was confirmed by Western blotting (**Figure 10B**). Immunoreactive bands in the range of 110–130 kDa were detectable in human spermatozoa and in mouse brain, which served as the positive control.

We also tested the presence of Slo1 transcripts in human sperm cells. Indeed, both Slo1 α (*kcnma1*) and Slo1 β3 (*kcnmb3*) transcripts were amplified from the total RNA isolated from human ejaculated sperm (**Figure 11**). Interestingly, Slo1 was shown to have a decreased sensitivity to ChTX in the complex with different auxiliary subunits (**Xia et al., 1999**). For example, the presence of the β3 subunit in the Slo1 complex requires micromolar, but not nanomolar concentrations of ChTX to completely inhibit channel activity (**Xia et al., 1999**). Indeed, the β3 subunit of Slo1 was shown to be expressed in testis (**Uebele et al., 2000**), and we also found transcripts of β3 (*kcnmb3*) from sperm RNA (**Figure 11**, right panel). Therefore, a reduced sensitivity of human KSper to ChTX (nearly complete inhibition of activity was achieved only with 1 μM ChTX) is likely due to the presence of the β3 auxiliary subunit in human sperm.

## Discussion

Potassium channels are indispensable for sperm physiology and are essential for membrane hyperpolarization upon sperm capacitation—the final sperm maturation in the female reproductive tract. Capacitation is also associated with intracellular alkalinization, which, in turn, has been shown to activate the calcium channel CatSper in human spermatozoa and the potassium channel KSper in murine sperm (**Zeng et al., 1995**; **Arnoult et al., 1996**; **Kirichok et al., 2006**; **Navarro et al., 2007**; **Lishko et al., 2011**). In mouse sperm, K⁺ currents originate mainly from the Slo3 channel, which is alkalinization-activated, calcium-insensitive potassium channel (**Schreiber et al., 1998**; **Zhang et al., 2006a, 2006b**; **Navarro et al., 2007**; **Santi et al., 2010**; **Yang et al., 2011**; **Zeng et al., 2011**). The currents we recorded from human sperm, however, show very different properties. According to our data, KSper currents

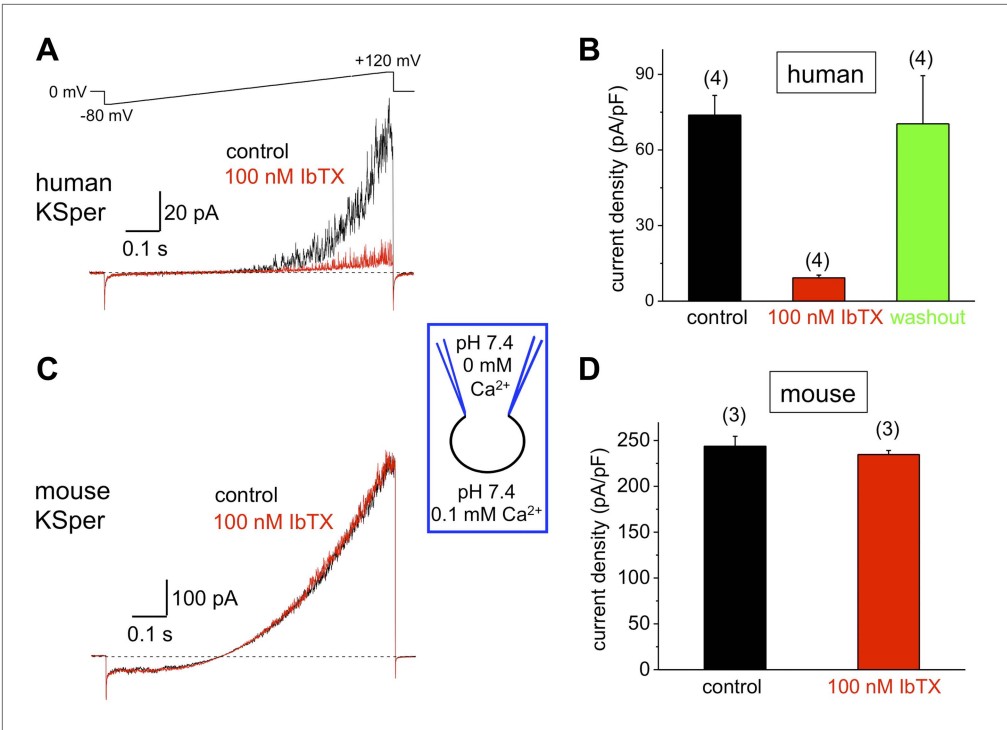

**Figure 6**. Human, but not mouse KSper is sensitive to the Slo1 channel blocker iberiotoxin (IbTX). (**A**) Representative human $I_{KSper}$ traces under control conditions (black) and in the presence of 100 nM IbTX (red) elicited in response to the shown voltage ramp. (**B**) Mean current densities (CDs) ± SEM calculated at +120 mV. CDs (human) were 74 ± 8 pA/pF (control), 9 ± 1 pA/pF (IbTX), and 70 ± 19 pA/pF (washout). (**C**) Representative mouse $I_{KSper}$ traces under control conditions (black) and in the presence of 100 nM IbTX (red) elicited in response to the voltage ramp as in (**A**). (**D**) CDs (mouse) were 244 ± 11 pA/pF (control) and 235 ± 4 pA/pF (IbTX). (n), number of experiments. Four human and three mouse sperm cells were used.

recorded either from human epididymal or ejaculated spermatozoa were alkalization-independent producing the same current amplitudes at $pH_i$ = 5.5 and 7.4. However, we found that human KSper instead is sensitive to intracellular calcium.

Capacitation also results in the elevation of intracellular calcium (*Visconti et al., 2002*). The C-terminus of Slo1 potassium channel possesses RCK domains with high-affinity $Ca^{2+}$ binding sites (*Moss et al., 1996*; *Schreiber and Salkoff, 1997*; *Jiang et al., 2001*; *Yuan et al., 2010*). This raises the possibility that Slo1 may represent human KSper. Associated γ- (leucine-rich repeat-containing proteins, LRRCs) and β-subunits further modulate channel behavior in response to calcium. Subunits γ1 (LRRC26), γ2 (LRRC52), γ3 (LRRC55), and γ4 (LRRC38) produce a shift towards more hyperpolarized membrane potentials, even in the absence of intracellular calcium and transcripts for all four subunits are detectable in human testis (*Yan and Aldrich, 2010*, *2012*; *Yang et al., 2011*). So far, four β subunits (β1–4) have been identified and are expressed in a tissue-specific manner. Subunits β2–4 are mainly expressed in brain and neurons, β3 is also detectable in testis, whereas the β1 subunit is preferentially found in smooth muscle cells (*Knaus et al., 1994b*; *Behrens et al., 2000*; *Brenner et al., 2000*; *Uebele et al., 2000*). hSlo1 activation time is reduced when the α subunit is co-expressed with β3 (*Brenner et al., 2000*). Inward currents with increased concentrations of calcium (10, 60, and 300 μM) occur when the α-subunit is expressed alone and are potentiated in the presence of subunit β1 and β3 (*Xia et al., 2000*). Keeping the $[Ca^{2+}]_i$ = 10 μM, an inward current becomes apparent when Slo1 α is co-expressed with β1 and β4, whereas β2 and β3 show no effect (*Brenner et al., 2000*; *Lippiat et al., 2003*). These data can be explained by the presence of β3 splice variants (β3a–d) arising from four different exons with each of them encoding for an alternative N terminus (*Zeng et al., 2008*). One study so far showed that β3b, β3c, and β3d transcripts are present in human testis, with β3d showing the greatest expression (*Uebele et al., 2000*). It is possible that $I_{KSper}$ of human spermatozoa originates from Slo1 α-subunits,

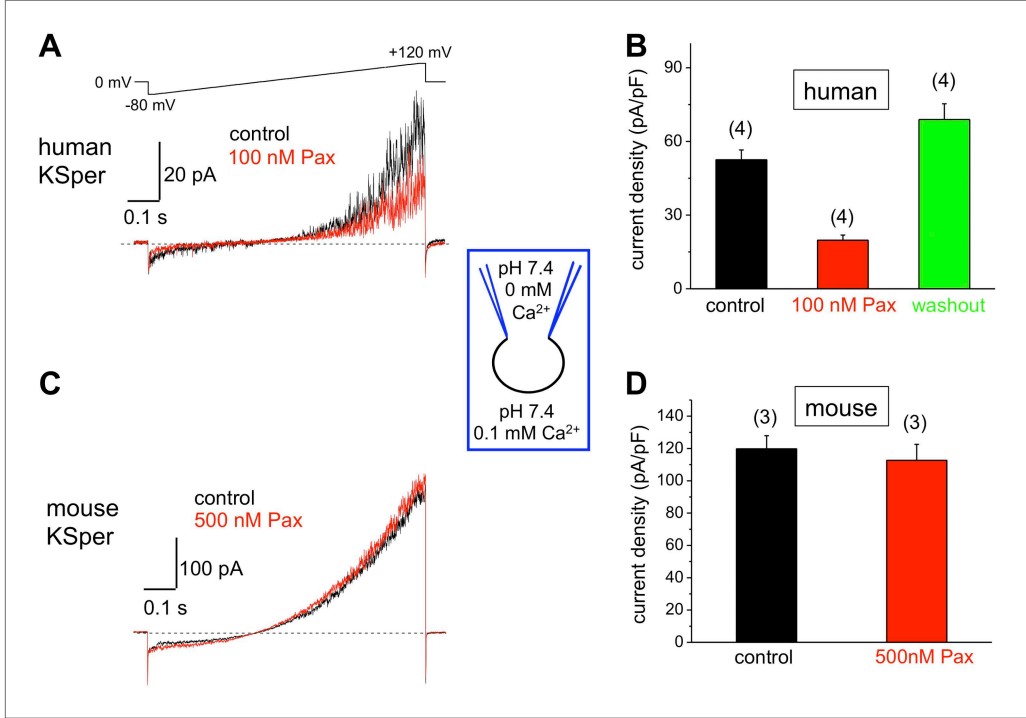

**Figure 7**. Human, but not mouse KSper is sensitive to the Slo1 channel blocker paxilline (Pax). (**A**) Representative human $I_{KSper}$ traces under control conditions and in the presence of paxilline elicited in response to the indicated voltage ramp. (**B**) Mean current densities (CDs) ± SEM calculated at +120 mV. Cells from three donors were used. CDs (human) were: 53 ± 4 pA (control), 20 ± 2 pA (100 nM paxilline), and 69 ± 6 pA (washout). (**C**) Representative mouse $I_{KSper}$ traces under control conditions and in the presence of paxilline elicited in response to the voltage ramp as in (**A**). (**D**) CDs (mouse) were: 119 ± 5 pA/pF (control) and 113 ± 10 pA/pF (500 nM paxilline). (n), number of experiments. Four human and three mouse sperm cells were used.

which are in the complex with γ, β, or even both auxiliary subunits. Indeed, according to our data, β3 transcripts are present in the RNA pool isolated from mature ejaculated human sperm cells. Moreover, our data show that elevated intracellular calcium strongly potentiates outward current and results in the appearance of an inward current. These hKSper properties (calcium sensitivity and pH-insensitivity) favor the idea that Slo1 represents the potassium channel in human spermatozoa.

Slo1 channels are potently blocked by the scorpion peptide toxins charybdotoxin (ChTX) (*Miller et al., 1985*; *Anderson et al., 1988*; *MacKinnon and Miller, 1988*) and iberiotoxin (IbTX) (*Galvez et al., 1990*; *Candia et al., 1992*; *Giangiacomo et al., 1992*). However, there is evidence that all four β-subunits can confer resistance to these toxins. It has been reported that IbTX effectively blocks recombinant hSloα with an $IC_{50}$ of 33 nM, but IbTX inhibition of recombinant hSloα in a complex with β1 raises $IC_{50}$ to 371 nM (*Lippiat et al., 2003*). Also, subunit β2 greatly reduces the sensitivity of the α subunit to ChTX ($IC_{50}$ = 1 nM vs 58 nM) (*Wallner et al., 1999*). Another study revealed that 20 nM ChTX was sufficient to block recombinant hSloα, whereas 100 nM of toxin was required to inhibit hSloα + β1. Moreover, even 100 nM of ChTX was insufficient to effectively block hSloα + β3 (*Xia et al., 1999*). Furthermore, slower blocking kinetics for ChTX and IbTX have been shown in hSloα + β4 constructs (*Meera et al., 2000*). Northern blot analyses demonstrate that mRNA for subunits β3 and β4 is detectable in human testis (*Brenner et al., 2000*), and it has been shown that the resistance to IbTX and ChTX is determined by the large extracellular loop of the β4 subunit (*Meera et al., 2000*). Since we observed different blocking kinetics with ChTX (93% reduction with 1 μM ChTX) and IbTX (87% reduction with 100 nM IbTX), it seems likely that in human sperm, the Slo1 channel is associated with β subunits that modulate channel behavior in response to these toxins.

Paxilline, a fungal indole alkaloid, has also been shown to inhibit Slo1 channels (*Knaus et al., 1994c*; *Sanchez and McManus, 1996*). Interestingly, Slo3 is paxilline insensitive, and recently it has been

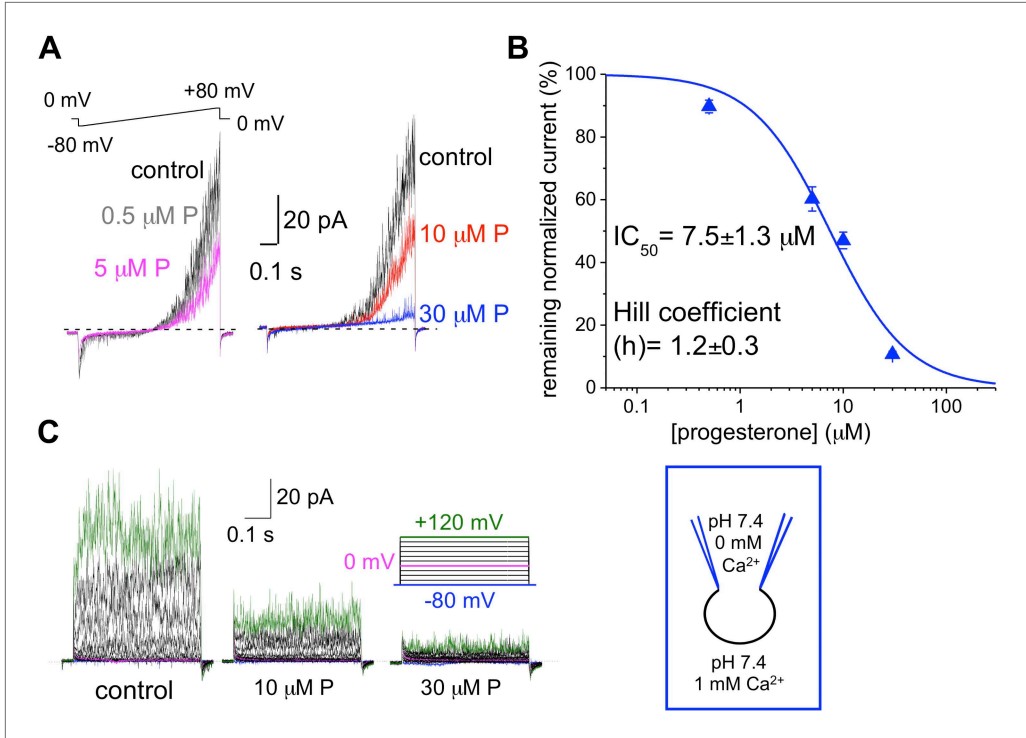

**Figure 8**. hKSper is blocked dose-dependently by progesterone (P). (**A**) Representative $I_{KSper}$ recordings from two sperm cells (left and right panel) in response to the given voltage ramp protocol under control conditions (black), 0.5 µM P (gray), 5 µM P (magenta), 10 µM P (red), and 30 µM P (blue). (**B**) Dose-dependent inhibition of human $I_{KSper}$ by progesterone. Human $I_{KSper}$ amplitudes were acquired at +80 mV at the end of the voltage ramps, as shown in (**A**). Current amplitudes in the presence of indicated progesterone concentrations were normalized onto control amplitudes (in the absence of progesterone). Remaining $I_{KSper}$ in the presence of 0.5 µM, 5 µM, 10 µM and 30 µM of P was: 90 ± 2%, 60 ± 4%, 47 ± 3% and 11 ± 1%, respectively. Data were fitted with the Hill equation. Data shown are means ± SEM of 4–10 sperm cells from three different donors. (**C**) Representative $I_{KSper}$ traces elicited by the given voltage step protocol of the control (left panel) and in the presence of 10 µM P (middle panel) and 30 µM P (right panel). Recordings are from the same cell as in (**A**).

demonstrated that paxilline binds to a glycine residue at position 311 within the S6 segment of the α-subunit (*Zhou et al., 2010*). This effect is independent of auxiliary subunits. However, the same study elucidated other factors within the Slo channel structure, which seem to be important for paxilline block. First, the turret region could determine the effectiveness of paxilline block. The turret is the extracellular loop between S5 and the pore domain, which contains more residues in Slo channels as compared to other K[+] channels (*Carvacho et al., 2008*; *Giangiacomo et al., 2008*; *Latorre et al., 2010*). Replacing the first half of the mSlo1 pore loop with the corresponding mSlo3 sequence leads to a five times greater paxilline inhibition and increases inhibition and washout rates (*Zhou et al., 2010*). A second source for altered abilities of paxilline inhibition is the pore loop region in the S6 segment, which differs in 10 residues between Slo1 and Slo3. When we applied 100 nM paxilline to human spermatozoa, only 62% of hKSper reduction was observed, whereas even a fivefold higher concentration did not affect currents recorded from mouse sperm. From these data, we conclude that in human spermatozoa, either the turret region or the S6 segment of BK channels show properties that do not allow a complete block of the K[+] currents by paxilline.

We and others previously have shown that progesterone is a potent non-genomic activator of CatSper with an $EC_{50}$ of 8 nM (*Lishko et al., 2011*; *Strünker et al., 2011*). But as apparent from this study, progesterone also blocks human KSper with an $IC_{50}$ of around 8 µM. Moreover, murine KSper is not affected by 10 µM progesterone. Together, the data from steroid and toxin treatment indicate that pharmacological properties of human and mouse KSper channels are quite different.

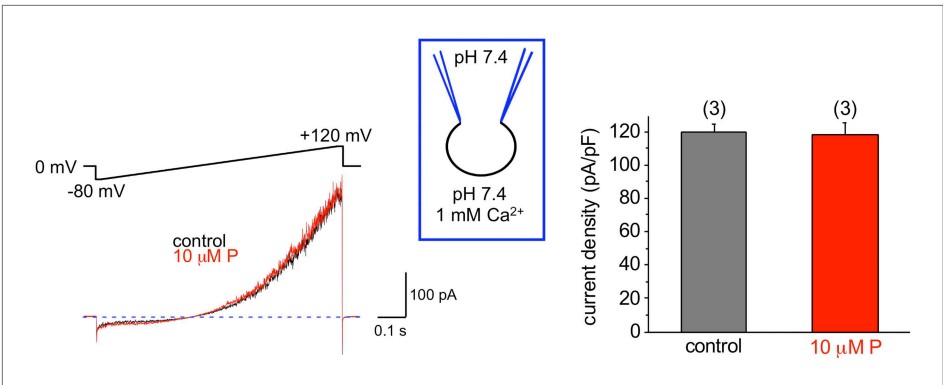

**Figure 9**. Mouse KSper is insensitive to progesterone (P). The left panel shows representative traces of mouse $I_{KSper}$ of the control (black) and in the presence of 10 μM P (red). The right panel shows current densities (CDs) acquired at +120 mV presented as mean ± SEM. CDs were: 119 ± 5 pA/pF (control) and 118 ± 8 pA/pF (10 μM P). (n), number of experiments. Three sperm cells were used.

In conclusion, we show that human $I_{KSper}$ originated from the sperm flagellum, the same compartment where also CatSper and Hv1 channels reside (*Lishko et al., 2010*, *2011*). Human KSper is a pH-independent, calcium-sensitive potassium channel sensitive to selective Slo1 inhibitors, such as charybdotoxin, iberiotoxin and paxilline, and is inhibited by micromolar concentrations of progesterone. Apart from its localization in sperm flagella, mouse KSper lacks all earlier-mentioned properties. Taken together, these results indicate that the human sperm potassium channel comprises the Slo1 protein and not Slo3. In addition, we propose the following model: the functional proximity of KSper to other sperm ion channels helps temporally coordinate their actions in a concerted manner during capacitation (*Figure 12*). In the uterus and the Fallopian tube, intracellular alkalinization is evoked by Hv1, thus activating CatSper channels. However, CatSper will not be fully active, as hKSper channels function as feedback regulators in response to calcium influx, thus retaining the membrane potential in a hyperpolarized state. In close proximity to the oocyte however, sperm encounter high concentrations of progesterone, which, in turn, will block hKSper, leading to membrane depolarization opening CatSper channels, which will become fully potentiated by the presence of progesterone. These events will lead to elevated levels of intracellular calcium in sperm, thereby initiating calcium-dependent processes such as hyperactivity and the acrosome reaction making the fertilization event possible.

## Materials and methods

### Human sperm cells

A total of 19 healthy fertile volunteers aged 21–38 years were recruited for this study. The study was conducted with approval of the Committee on Human Research at the University of California, Berkeley (protocol 10-01747, IRB reliance #151), and University of California, San

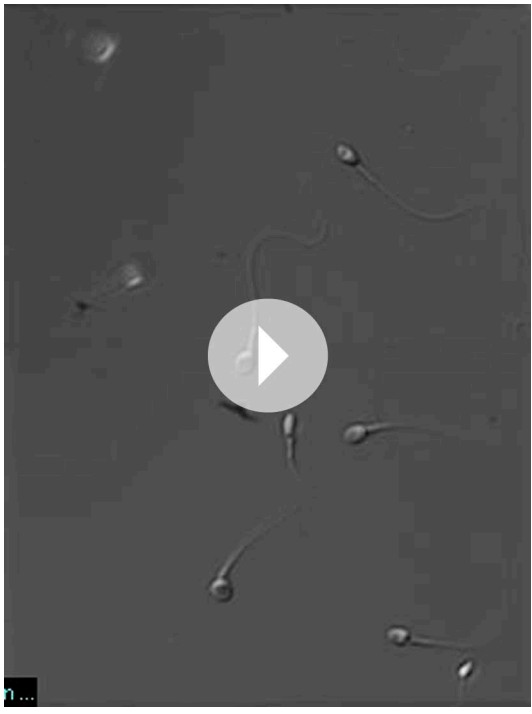

**Video 1**. Inhibition of hKSper induces a hyperactivation-like motility pattern. Normal motility of human spermatozoa in the control HS solution. Scale bar is 5 mm. Recording was slowed down five times.

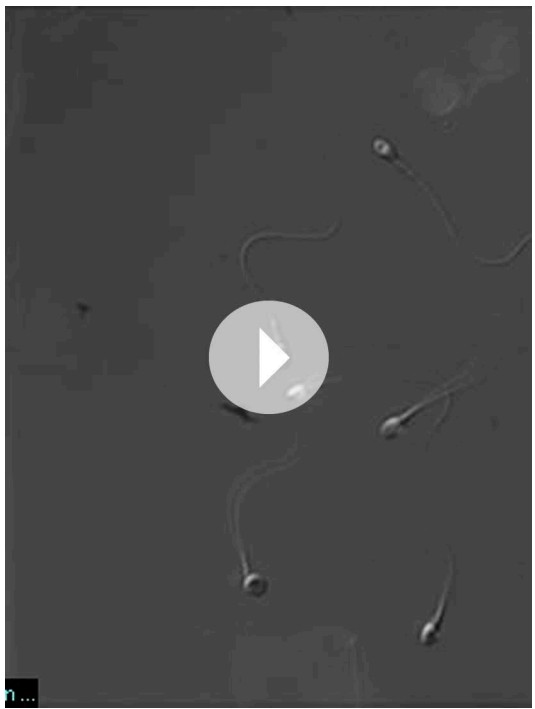

**Video 2**. Inhibition of hKSper induces a hyperactivation-like motility pattern. Motility of human spermatozoa is altered after incubation in HS solution, which contained 100 nM of charybdotoxin (ChTX). Scale bar is 5 mm. Recording was slowed down five times.

Francisco (protocol 10-04868). Informed consent was obtained from all participants. Ejaculates were obtained by masturbation and spermatozoa were purified following the swim-up protocol as previously described (*Lishko et al., 2011*). Men with proven fertility who were undergoing sperm retrieval procedures or a vasectomy reversal in the UCSF Center for Reproductive Health were also included in this study. As part of the ongoing IRB-approved LIFE (Lifestyle, Fertility, and Evaluation) study, men who agreed to participate donated portions of surgical specimens. All men enrolled in the present study had a documented history of prior paternity and had undergone a vasectomy in the past. As part of routine clinical care, these men elected to undergo a sperm retrieval procedure (microscopic epididymal sperm aspiration, MESA, or percutaneous epididymal sperm aspiration, PESA) combined with in vitro fertilization (IVF) or a vasectomy reversal. An aliquot of epididymal fluid was used for the present study with patient consent.

## Animals

Male C57BL/6 mice were purchased from Harlan Laboratories (Livermore, CA) and were kept in the Animal Facility of the University of California, Berkeley. All experiments were performed in strict accordance with the NIH Guidelines for Animal Research and approved by UC Berkeley Animal Care and Use Committee, the approved protocol MAUP #R352-012. Animals were killed by $CO_2$ asphyxiation and cervical dislocation, and sperm were collected as described previously (*Wennemuth et al., 2003*).

## Reagents

Progesterone was purchased from CalBiochem (EMD Millipore, Darmstadt, Germany), charybdotoxin and iberiotoxin from Tocris Bioscience (Bristol, UK), and all other compounds were obtained from Sigma (St. Louis, MO, USA).

## Electrophysiology

Gigaohm seals were formed at the cytoplasmic droplet (*Cooper, 2011*) of highly motile cells or separated flagella in standard high saline (HS) buffer containing (in mM) 130 NaCl, 20 HEPES, 10 lactic acid, 5 glucose, 5 KCl, 2 $CaCl_2$, 1 $MgSO_4$, 1 sodium pyruvate, pH 7.4 adjusted with NaOH, 320 mOsm/l as reported in *Lishko et al. (2010, 2013)*. Transition into whole-cell mode was achieved by applying voltage pulses (499–611 mV, 1 ms) and simultaneous suction. Cells were stimulated every 5 s, data were sampled at 10 kHz and filtered at 1 kHz and access resistance was 21–57 MΩ. Pipettes (13–16 MΩ) were filled with 130 mM $KMeSO_3$, 20 mM HEPES, 4 mM KCl, 10 mM EGTA, 1 mM EDTA, and pH 7.4 was adjusted with KOH, 330 mOsm/l. In the experiments with $NH_4Cl$, pipette solutions were of similar composition, but contained just 5 mM HEPES to allow efficient intracellular pH changes. The nominal free bath solution (NMF) consisted of (in mM) 140 $KMeSO_3$, 20 HEPES, and pH 7.4 was adjusted with KOH, 320 mOsm/l. To inhibit monovalent currents through CatSper channels (*Smith et al., 2013*), 0.1–1 mM $Ca^{2+}$ was added to the NMF solution, as indicated. To elicit potassium currents through CatSper, currents were recorded in a $K^+$-based divalent free bath solution (K-DVF) containing (in mM) 140 $KMeSO_3$, 45 HEPES, 1 EDTA, 7.4 adjusted with KOH, 320 mOsm/l. Inside (pipette) solutions with different concentrations of free $Ca^{2+}$ contained (in mM) 130 $KMeSO_3$, 20 HEPES, 4 KCl and 1 BAPTA, 1 EDTA, 1 EGTA (for 100 nM $Ca^{2+}$) or 1 HEDTA (for 50 µM $Ca^{2+}$), respectively. $CaCl_2$ was added according to WinMAXC32 version 2.51 (Chris Patton, Stanford University). Since changing of

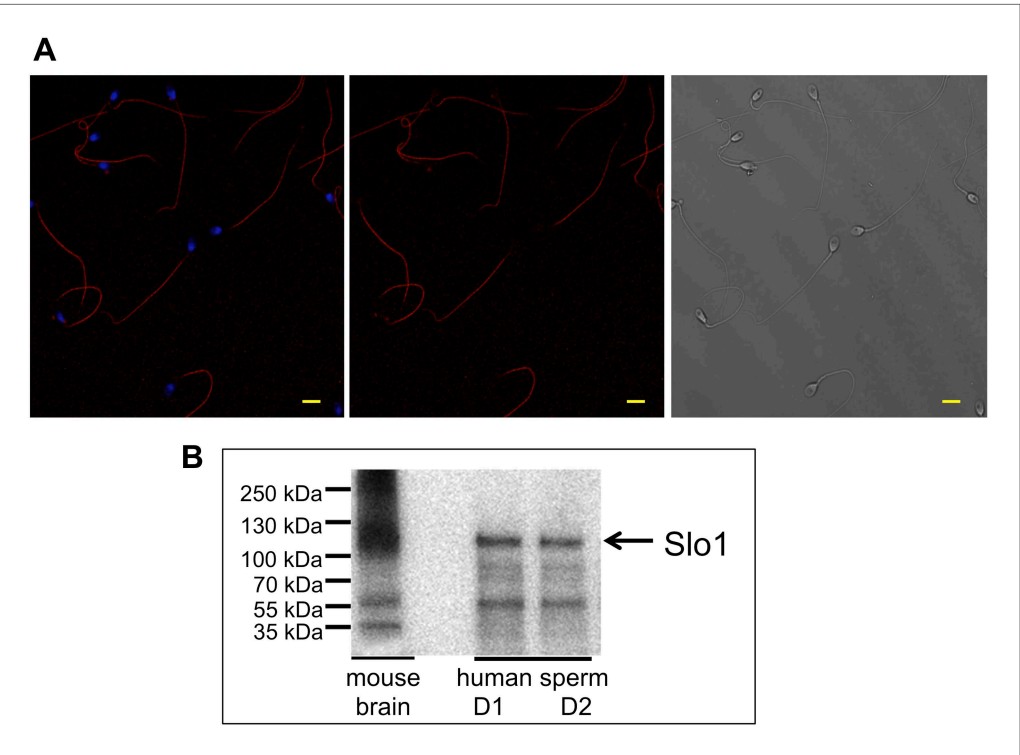

**Figure 10**. Slo1 protein is present in human spermatozoa. (**A**) Human sperm immunostaining with primary polyclonal anti-Slo1 antibodies and Cy3-conjugated secondary antibodies. Left and middle panels show Slo1 staining localized to the principal piece of human sperm flagellum. Left panel: nuclei are stained by DAPI. Right panel: DIC image of the same cells. Scale bar is 5 mm. (**B**) Representative immunoblot of the mouse brain (positive control) and human spermatozoa from two different donors (donor 1 and donor 2: D1 and D2, respectively).

the pipette solution cannot be easily done on one cell, the data obtained with different intracellular pH or different intracellular $[Ca^{2+}]$ are a combination of recordings from multiple cells. However, since the changing of bath solution can be easily accomplished on the same cell, the experiments with different bath solutions (addition of EDTA, extracellular calcium, $NH_4Cl$, ChTX, IbTX, Paxilline, progesterone, etc) were performed on the same sperm cell (flagellum): before and after addition of the above-mentioned compound. Data were analyzed with Clampfit 10.3 (Molecular Devices, Sunnyvale, CA, USA) and OriginPro 8.6 (OriginLab Corp., Northampton, MA, USA). Statistical data are presented

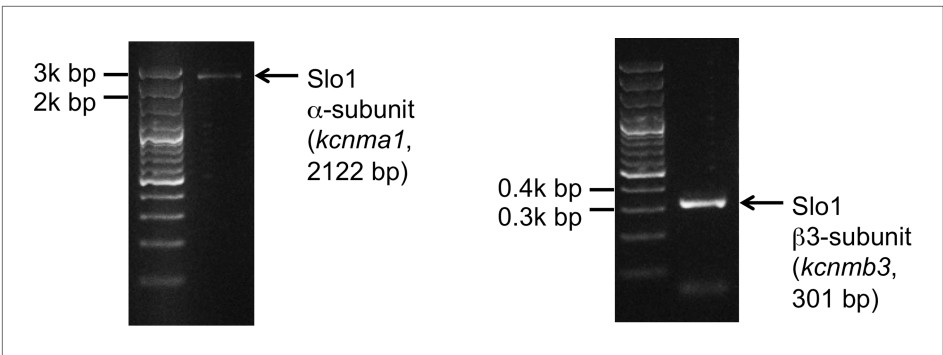

**Figure 11**. Slo1 transcripts are present in human spermatozoa. PCR bands of the portion of the translated region of *kcnma1* (left panel; 1433–3554 bp, corresponding to the coding sequence of splice isoform1; UniProt # Q12791), and of the translated region of *kcnmb3* (right panel; 529–829 bp of the coding sequence of splice isoform 3d, Uniprot # Q9NPA1).

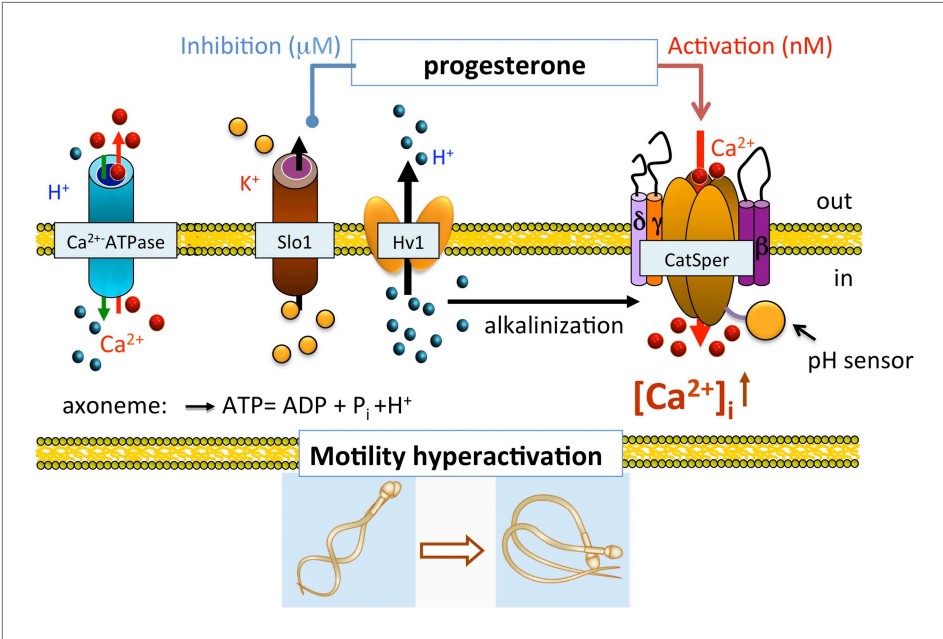

**Figure 12**. Role of human KSper (Slo1) in sperm physiology. In the uterus and fallopian tube, CatSper is partially activated due to the intracellular alkalinization evoked by proton extrusion through Hv1 and picomolar- to nanomolar progesterone (P) concentrations. However, to achieve full activation of CatSper, flagellar plasma membrane must be depolarized. This is achieved by the inhibition of sperm KSper, the channel responsible for membrane hyperpolarization. In close proximity to the oocyte, spermatozoa encounter micromolar concentrations of P, which inhibit hKSper, resulting in membrane depolarization. These events allow full activation of CatSper, trigger sperm hyperactivation, allow spermatozoa to penetrate through the egg protective vestment, and make fertilization possible.

as mean ± standard error of the mean (SEM), and n indicates number of experiments. All electrophysiology experiments were performed at 24°C.

## Immunocytochemistry

Cells were seeded onto cover slips in HS solution and allowed to adhere for 30 min at room temperature (RT). Cells were fixed with ice cold methanol for 1 min, washed in PBS, and subsequently permeabilized with PBS/0.1% Triton (PBS-T) with 5% BSA for 1 hr at room temperature. Incubation with the primary antibody (rabbit anti-Maxi K+ alpha, 1:100 in PBS-T and BSA, Thermo Scientific, # PA1-923) was performed at 4°C overnight. Cells were then washed in PBS-T and incubated with Cy3 conjugated goat anti-rabbit IgG for 45 min at room temperature. After washing, the samples were mounted with ProLong Gold antifade reagent. Images were taken on a Zeiss LSM 710 microscope (Carl Zeiss Microscopy, Oberkochen, Germany) and processed with the Zeiss ZEN 2010B imaging software.

## Electrophoresis and immunoblotting

Purified human spermatozoa were centrifuged and re-suspended in lysis buffer containing 0.1% SDS, 0.5% sodium deoxycholate, 50 mM dithiothreitol (DTT), 20 mM EDTA, 4 M urea and protease inhibitor cocktail (Roche). After sonication for 5 min, the cell suspension was mixed 1:0.5 with water, then mixed 1:1 with sample buffer, and sonicated again for 5 min β-mercaptoethanol (5%) was added to each sample and after boiling (5 min, 100°C) samples were transferred to a 4–12% polyacrylamide gel and blotted on PVDF membranes. Membranes were fixed with methanol followed by blocking with 3% IgG-free BSA in PBS containing 0.1% Tween (PBS-T) for 30 min at room temperature. Subsequently, membranes were incubated with 1 μg of mAb anti-Slo1 (clone L6/60) (UC Davis/NIH NeuroMab Facility, Davis, CA, USA) overnight at 4°C. After washing three times with PBS-T, the membranes were incubated with HRP-conjugated anti-mouse IgG (1:20,000 dilution) for 1 hr at room temperature. Protein bands were detected by enhanced chemiluminescence on a Fluor Chem M imaging system (Protein Simple).

## RNA extraction and RT-PCR

Spermatozoa were purified by swim-up procedure and total donor-specific RNA was extracted from purified spermatozoa using a Qiagen RNeasy mini kit followed by cDNA synthesis with a Phusion RT-PCR kit (Finnzymes, MA, USA). The donor- specific translated regions of kcnma1 between 1433 bp and 3554 bp (corresponding to the canonical coding sequence of Slo1 α isoform1; UniProt id Q12791) and kcnmb3 between 529 bp and 829 bp of the canonical coding sequence (Slo1β iso-form 3d, Uniprot id Q9NPA1) were amplified using the following primers: 5′-ATGCCTCGAAT ATCATGAGAG-3′ (kcnma1, forward), 5′-TATATTGGTTGATCTGGTTAGCC-3′ (kcnma1, reverse); 5′-CTCGCCTAGGTTCTTCGATCACAAAAATGG-3′ (kcnmb3, forward), and a reverse 5′-ATCGCTCG AGCTGCTCTTCCTTTGCTCCT-3′ (kcnmb3, reverse). All PCRs were carried out for 40 cycles of replication and had annealing temperatures of 61°C. The obtained PCR products were gel-purified and sequence-verified (Sequetech, Mountain View, CA, USA).

## Video recording of human sperm movement

Purified spermatozoa were plated onto 5-mm coverslips in HS solution. Sperm movement was recorded within the first 3 hr after sperm retrieval with a high speed GX-1 Memrecam camera (NAC Image Technology) attached to an Olympus IX71 microscope (Olympus Corp., Central Valley, PA, USA). The recording speed was 960 frames per second (fps), and videos were slowed down to playback at 200 fps where indicated.

## Acknowledgements

We thank Sam Coleman from the Molecular Imaging Center, UC Berkeley, for his help with the confocal images. We thank Dr Yuriy Kirichok from UCSF for the help and advice with the pilot experiments, and Dr Donner Babcock and Dr Melissa Miller for helpful suggestions. We are also very grateful to DO Nors for continuing invaluable contributions to this research. This work was supported by Winkler Family Foundation Fellowship and #5-FY13-204 Basil O'Connor March of Dimes award to PVL.

## Additional information

### Funding

| Funder | Grant reference number | Author |
|---|---|---|
| Basil O'Connor March of Dimes | #5-FY13-204 | Polina V Lishko |
| Winkler Family Foundation | | Polina V Lishko |
| National Institute of Child Health and Human Development, National Institutes of Health | K12 | James F Smith |

The funders had no role in study design, data collection and interpretation, or the decision to submit the work for publication.

### Author contributions

NM, Helped design the experiments, performed most of the experiments, analyzed and interpreted data, and wrote the manuscript; NMN, Performed molecular biology experiments, analyzed and interpreted data, and helped write the manuscript; S-ASC, Performed immunocytochemistry experiments, analyzed the data, and helped revise the manuscript; JFS, Contacted patients, conducted surgeries and provided biopsy samples, discussed the results, and commented on the manuscript; PVL, Conceived the project, designed the experiments, performed pilot experiments, analyzed and interpreted data, and wrote the manuscript

### Ethics

Human subjects: The study was conducted with approval of the Committee on Human Research at the University of California, Berkeley (protocol 10-01747, IRB reliance #151), and University of California, San Francisco (protocol 10-04868). Informed consent was obtained from all participants. Men with proven fertility who were undergoing sperm retrieval procedures or a vasectomy reversal

in the UCSF Center for Reproductive Health were included in this study. As part of the ongoing IRB-approved study (approval number 10-04868), men who agreed to participate donated portions of surgical specimens.

Animal experimentation: Male C57BL/6 mice were purchased from Harlan Laboratories (Livermore, CA) and were kept in the Animal Facility of the University of California, Berkeley. All experiments were performed in strict accordance with the NIH Guidelines for Animal Research and approved by UC Berkeley Animal Care and Use Committee, the approved protocol MAUP #R352-012. Animals were killed by $CO_2$ asphyxiation and cervical dislocation.

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
