## [Decision Letter]

Thank you for sending your work entitled “Slo1 is the principal potassium channel of human spermatozoa” for consideration at *eLife*. Your article has been favorably evaluated by a Senior editor and 3 reviewers, one of whom is a member of our Board of Reviewing Editors.

The following individuals responsible for the peer review of your submission have agreed to reveal their identity: Richard Aldrich (Reviewing editor), David Clapham, and Dejian Ren (peer reviewers).

The Reviewing editor and the other reviewers discussed their comments before we reached this decision, and the Reviewing editor has assembled the following comments to help you prepare a revised submission.

Mannowetz et al. studied the K^+^ conductance in human sperm (hKsper channel) with whole sperm cell patch clamp recordings. Along with Ca^2+^ and H^+^ channels, K^+^ channels are fundamental in sperm function in setting the resting membrane potential and in controlling Ca^2+^ influx. In mouse sperm, it has been redundantly shown that the major K^+^ channel (mKsper) is formed by the pH-sensitive mSlo3 (48; 78). Using whole cell patch recording from human sperm and various channel inhibitors, the authors surprise us with the finding that the major K^+^ conductance in human sperm is not formed by Slo3 but is likely formed by Slo1, a K^+^ channel with biophysical properties and regulation quite different from those of Slo3.

The evidence supporting the conclusion that hKsper is unlikely formed by Slo3 is quite convincing. They include the hKsper's insensitivity to intracellular alkalinization (a Slo3 “activator”) and its sensitivity to Slo1 blockers. The authors also provide evidences supporting that hKsper is formed by Slo1. They include hKsper's sensitivity to [Ca^2+^]i and Slo1 inhibitors, and the presence of Slo1 proteins in sperm tail.

In addition, the authors discovered an inhibition of hKsper by progesterone at physiological concentration of ∼ μM. While the mechanism underlying such inhibition needs further studies in the future, the findings provide a quite novel mechanism for the action of progesterone: the hormone induces Ca^2+^ influx through CatSper by directly activating the CatSper channel (37; 60) and by, perhaps indirectly, inhibiting hKsper and promoting sperm depolarization.

In summary, the studies uncover a very important component in sperm ionic conductance. Together with previous findings from several labs including the authors', the studies presented in the manuscript complete the identification of all the major ion channels important for sperm function. The findings thus represent a major step forward in our understanding of human fertilization.

The following issues need to be resolved for the paper to be acceptable:

1) There is no indication as to whether some of the experiments were done on the same cell, or flagellum, before and after treatment, or whether all of the comparisons are between different cells under differ conditions. This should be made explicitly clear. Demonstration of pharmacological effects on a single preparation would strengthen the results.

2) Is there an independent measurement that the NH_4_Cl treatment actually causes internal alkalinization? If this has been worked out in previous literature it should be referenced.

3) The experiments showing lack of paxilline sensitivity of mouse KSPER do not provide compelling evidence that the channels are not Slo1/Slo3 heteromers without also showing that such heteromers are in fact paxilline sensitive. The implicit assumption that any contribution of Slo1 to the heteromer would confer sensitivity is unsupported. The claims about heteromers should be deleted unless such evidence is also presented. The presented results do however provide further evidence that human and mouse sperm potassium currents are indeed different, and should not be deleted.

4) Time constant fits – it is worrisome that the fitted time constants are in the time range where the current time courses were extrapolated to the origin from later time points. While the general point of acceleration at higher calcium concentrations is reasonable, the time constant values are highly suspect.

5) G/V cures were calculated from tail current amplitude, but the tail currents presented do not inspire much confidence that they are well resolved. This could be fixed by providing a higher gain and expanded time scale for tail currents to allow assessment of data quality.

6) While the differences between human and mouse K currents are supported, it is disconcerting that the mouse currents have such higher amplitudes. Is there some technical reason for this based on cell size differences etc? Are there physiological reasons that make sense?

7) Figure 4 shows only currents from voltage steps. Voltage ramps, as in the other figures, would be preferable.

8) The authors should consider showing more comparisons between human and mouse results, as in Figure 4. While the mouse results are known from the literature, comparisons in the figures would help illustrate the differences, which are the main point of the paper.

---

## [Author Response]

*1) There is no indication as to whether some of the experiments were done on the same cell, or flagellum, before and after treatment or whether all of the comparisons are between different cells under differ conditions. This should be made explicitly clear. Demonstration of pharmacological effects on a single preparation would strengthen the results*.

We have clarified this by adding a sentence in the corresponding figure legends and by stating which type of preparation (whole sperm cell vs flagellum were used). In fact, the experiments with sperm flagella were only included in Figure 1. All other experimental data were acquired from whole sperm cells. Moreover, since changing of the pipette solution cannot be easily done on one cell, the data obtained with different intracellular pH or different intracellular [Ca^2+^] are a combination of recordings from multiple cells. However, since the changing of bath solution can be easily accomplished on the same cell, the experiments with different bath solutions (addition to EDTA, NH_4_Cl, ChTX, IbTX, Paxilline, progesterone, extracellular calcium, etc) were performed on the same sperm cell (or the same flagellum), before and after addition of the mentioned above compound. We have added this clarification to the Methods section.

*2) Is there an independent measurement that the NH*_*4*_*Cl treatment actually causes internal alkalinization? If this has been worked out in previous literature it should be referenced*.

Yes, addition of NH_4_Cl to the bath solution is a standard technique to effectively and quickly raise an intracellular pH. We have mentioned this in the text and have cited the previous literature. Moreover, as indicated on Figure 2 (right panels), addition of 10 mM NH_4_Cl strongly increases potassium efflux under divalent free conditions (DVF). This happens because in DVF conditions most of the potassium efflux is carried out by pH-sensitive CatSper channel. The fact that K^+^ efflux is strongly up-regulated upon addition of NH_4_Cl in DVF, indicates that intracellular alkalinization has been achieved, and CatSper was potentiated (as it should). Recording in DVF conditions served as an intrinsic control in order to verify the appearance of intracellular alkalinization.

*3) The experiments showing lack of paxilline sensitivity of mouse KSPER do not provide compelling evidence that the channels are not Slo1/Slo3 heteromers without also showing that such heteromers are in fact paxilline sensitive. The implicit assumption that any contribution of Slo1 to the heteromer would confer sensitivity is unsupported. The claims about heteromers should be deleted unless such evidence is also presented. The presented results do however provide further evidence that human and mouse sperm potassium currents are indeed different, and should not be deleted*.

We agree with this statement and have removed the speculation about Slo1/Slo3 heteromer presence. Moreover, we have performed additional experiments with Slo1 specific inhibitor and a scorpion toxin – iberiotoxin (IbTX) – on both human and mouse sperm (Figure 4). As expected, just 100 nM of IbTX eliminated 87% of the human KSper, while no effect on mouse KSper was observed. These experiments provide the strongest proof that Slo1 protein constitutes the main potassium channel of human spermatozoa.

*4) Time constant fits – it is worrisome that the fitted time constants are in the time range where the current time courses were extrapolated to the origin from later time points. While the general point of acceleration at higher calcium concentrations is reasonable, the time constant values are highly suspect*.

Intracellular calcium notably accelerates human KSper. Regretfully, the fast channel kinetics produced a situation where the beginning of the channel opening partially overlapped with capacitance artifacts therefore making quantitative measurements of the activation time constantly problematic. We agree with the reviewers on this point and removed time constant measurements. We have referred to this phenomenon in the revised manuscript.

*5) G/V cures were calculated from tail current amplitude, but the tail currents presented do not inspire much confidence that they are well resolved. This could be fixed by providing a higher gain and expanded time scale for tail currents to allow assessment of data quality*.

We are thankful for this suggestion. The closer examination of the tail currents revealed that we are dealing with a similar problem as with measurements of activation time constants. The closing of human KSper produced fast tail currents, especially fast in the presence of intracellular calcium. Such tail currents significantly overlapped with capacitance artifacts (the duration of an artifact was ∼10 ms, while tail currents were essentially irresolvable after 8 ms). Therefore, we decided to remove the G/V curve data from the manuscript.

*6) While the differences between human and mouse K currents are supported, it is disconcerting that the mouse currents have such higher amplitudes. Is there some technical reason for this based on cell size differences etc? Are there physiological reasons that make sense*?

Indeed, mouse spermatozoa are twice larger than human sperm cells: human sperm capacitance is usually within 1 pF, while mouse is about 2.5 pF. However, the fact that the current densities are still larger (in fact, twice larger) in mouse sperm points to the potential differences in KSper expression. We have discussed this in the revised manuscript. In addition, recording from mouse spermatozoa were done under conditions where Slo3 is fully activated (pH= 7.4), while recording from human sperm was done at 0 intracellular calcium. The latter creates conditions for only partial Slo1 activation (human KSper in the presence of micromolar intracellular calcium are twice larger than in the absence). Another physiological explanation could be that mouse CatSper is operational under more negative membrane potentials (Vm) than human CatSper: V1/2 of human CatSper is +30mV (pH 7.5, +progesterone), while V1/2 of mouse CatSper is -11mV (pH 7.5, progesterone insensitive). Higher KSper expression in human sperm cells would further hyperpolarize the membrane essentially making it impossible for human CatSper to function. This does not apply to the mouse CatSper, however, as it is functional in more hyperpolarized conditions.

*7)*
Figure 4
*shows only currents from voltage steps. Voltage ramps, as in the other figures, would be preferable*.

We have replaced voltage step data with that of voltage ramps.

*8) The authors should consider showing more comparisons between human and mouse results, as in*
Figure 4*. While the mouse results are known from the literature, comparisons in the figures would help illustrate the differences, which are the main point of the paper*.

Indeed, such a comparison makes the data stronger. We have re-arranged the data to present the results in the manner of Figure 4. Our additional experiments with IbTX (almost complete inhibition of human KSper, while no effect on mouse KSper was observed (Figure 4)) also follow the same logic.